# Exploring Hospital Inpatients’ Awareness of Their Falls Risk: A Qualitative Exploratory Study

**DOI:** 10.3390/ijerph20010454

**Published:** 2022-12-27

**Authors:** Elissa Dabkowski, Simon J. Cooper, Jhodie R. Duncan, Karen Missen

**Affiliations:** 1Institute of Health and Wellbeing, Federation University Australia, Gippsland, VIC 3842, Australia; 2Health Innovation and Transformation Centre, Federation University Australia, Berwick, VIC 3806, Australia; 3Research Unit, Latrobe Regional Hospital, Traralgon, VIC 3844, Australia

**Keywords:** falls, fall prevention, adult, patient, perception, regional, rural

## Abstract

Patient falls in hospital may lead to physical, psychological, social and financial impacts. Understanding patients’ perceptions of their fall risk will help to direct fall prevention strategies and understand patient behaviours. The aim of this study was to explore the perceptions and experiences that influence a patient’s understanding of their fall risk in regional Australian hospitals. Semi-structured, individual interviews were conducted in wards across three Australian hospitals. Participants were aged 40 years and over, able to communicate in English and were mobile prior to hospital admission. Participants were excluded from the study if they returned a Standardised Mini-Mental State Examination (SMMSE) score of less than 18 when assessed by the researcher. A total of 18 participants with an average age of 69.8 years (SD ± 12.7, range 41 to 84 years) from three regional Victorian hospitals were interviewed for this study. Data were analysed using a reflexive thematic analysis identifying three major themes; (1) Environment (extrinsic) (2) Individual (intrinsic), and (3) Outcomes, as well as eight minor themes. Participants recognised the hazardous nature of a hospital and their personal responsibilities in staying safe. Falls education needs to be consistently delivered, with the focus on empowering the patient to help them adjust to changes in their clinical condition, whether temporary or permanent.

## 1. Introduction

Fall-related events are a significant source of harm and constitute the most frequently reported adverse event in hospitals [1]. Although falls can occur across the lifespan, people aged 65 years and older have the highest risk of falling in hospital, with risk increasing with advanced age [2]. Physical consequences that can result from a fall may significantly impact a person’s quality of life [3]. Likewise, there are financial impacts to organisations after an inpatient fall, including increased hospital length of stay and additional healthcare resources, e.g., personnel, equipment [4]. The majority of patient falls in hospital are avoidable and related to intrinsic risk-taking behaviours and extrinsic/environmental influences [5]. Despite extensive research and educational campaigns focused on falls prevention and management, falls in hospital are common and can have negative consequences [6]. Falls can occur in all environments, such as at home or in the community; however, there is an expectation that a safe and high-quality health system will keep patients safe from avoidable harm [7]. In an Australian context, there were approximately 47,551 falls in hospitals in 2020–2021 which resulted in patient harm [8]. Given that rates are predicted to increase because of an ageing global population, considerable efforts are needed to prevent hospital falls and to reduce strain on healthcare organisations [9].

The World Health Organisation [WHO] has outlined a systems-approach to address falls, in which interventions are categorised into three domains: safer people, safer environments and safer policies and legislation [10]. The ‘safer people’ domain includes the intent to increase falls awareness through education relating to an individuals’ vulnerability to falls and the risk factors associated with a fall [10]. For example, the risk of falling in hospital is likely to be increased due to the unfamiliar environment, co-morbidities, changes in medication, cognitive impairment or delirium and prolonged immobilisation leading to muscle atrophy [11,12,13]. However, studies show that many inpatients do not consider themselves to be at risk of falling in hospital and there is a disparity between their perceived fall risk and their clinical risk of falling [12,14,15]. The ‘safer environment’ domain signifies the elimination of falls hazards, whilst the ‘safer policies and legislation’ enforce change, such as minimum standards for falls risk assessments [10].

The recently updated World Falls Guidelines (WFG) strongly recommend that clinicians enquire about falls perceptions, causes, future risk and strategies for prevention in older adults [16]. Research has shown that reduced falls risk awareness, low self-efficacy and poor engagement with strategies can contribute to falls in hospitalised adults [17]. A common finding in both international and Australian literature is that many older adults do not consider themselves at risk of falling in hospital [12,14,15,18,19,20]. Therefore, understanding perception, as well as patient behaviours regarding fall risk, may help to focus fall prevention strategies in hospitals, especially patient and carer education [21]. Previous qualitative studies in an Australian hospital context have explored patients’ perception on falls education [12,22], their experiences of falling in hospital [23] and risk-taking behaviour in hospital [24]. Individualised falls education is advantageous to participants and increases motivation to engage with strategies [12,22]. There was an initial denial of clinical falls risk in participants, which transformed into acceptance as they regained their independence following a fall [23]. The findings from these studies are limited to private hospital settings [12], based on a specific hospital intervention [22] and relevant to one geographic metropolitan region [24].

In 2021–2022, most Australian inpatient falls occurred in public hospitals (5.5 falls per 1000 separations), compared to those in private hospitals (1.9 falls per 1000 separations) [8]. In the State of Victoria, Australia, hospital fall rates were higher in regional hospitals compared to those in metropolitan areas [8]. Studies have established that a large proportion of hospital falls are unwitnessed, with most inpatient falls occurring in the hospital bedroom [21,25,26,27]. This was also verified in an unpublished snapshot clinical audit from a regional Victorian public hospital reporting that 95% of fall events were unwitnessed by hospital staff. Given the disparity between patients’ perceived falls risk and their actual clinical risk [12,14,15], there is a clear need to explore this from a patient’s perspective, especially with the continued high number of unwitnessed falls. Falls risk awareness is associated with increased rehabilitation engagement and motivation [28], so exploring the perceptions of inpatients in this context will contribute to understanding and provide recommendations for falls prevention management for this cohort. 

### Aim

The overarching aim of this study is to explore the perceptions and experiences that influence a patient’s understanding of their falls risk in Australian rural and regional public hospitals. 

## 2. Design and Methods

A qualitative exploratory study was conducted to explore the patients’ falls risk awareness in rural and regional public hospitals. This study formed part of a larger mixed methods convergent design as described by Creswell and Plano Clark [29] to explore patients’ perceptions of their falls risk. The philosophical assumption underpinning this research is that of pragmatism in which multiple methods of data collection are utilised to inform the problem under study [29]. Both qualitative and quantitative data were collected simultaneously; however, this paper presents the findings of the qualitative component only. This study is reported in accordance with the Consolidated Criteria for Reporting Qualitative Research (COREQ) checklist (Appendix A) [30].

### 2.1. Ethical Considerations

High-risk ethical approval was granted from both Latrobe Regional Hospital Human Research Ethics Committee and Federation University Human Research Ethics Committee prior to participant recruitment (2022-02 HREA and 2022-057, respectively). Each participant received a plain language information statement (PLIS) and was advised that they could withdraw from the study at any time prior to data analysis. Both written and verbal consent was obtained from participants. For confidentiality purposes, participants were assigned codes.

### 2.2. Recruitment

Participants were recruited from acute medical, surgical, orthopaedic and rehabilitation wards across three rural/regional hospitals in the State of Victoria, Australia using a purposive sampling strategy. To participate in this study, inpatients were required to be aged ≥ 40 years, English-speaking and have ambulatory capacity prior to their admission. This age range was purposefully selected based upon retrospective clinical audit falls data from one of the participating regional hospitals, which established that falls were not limited to older adults. To determine cognitive status, participants were assessed by the researcher using a Standardised Mini-Mental State Examination (SMMSE) and were deemed eligible if they returned a score of ≥18. A score between 18 and 24 on the SMMSE can be used to indicate mild to moderate cognitive impairment [31]. People with cognitive deficits are often omitted from research [32] yet have an increased risk of falling in hospital [13]. For this reason, the researchers extended the eligibility criteria to include inpatients with mild to moderate cognitive impairment. The lead researcher liaised with the Assistant Nurse Unit Manager (ANUM) at the start of the shift to assist with recruitment and to avoid disrupting clinical management. The ANUM identified inpatients based on the eligibility criteria and used their clinical judgement to advise which patients were clinically unsuitable for the study, e.g., such as those who were unavailable due to COVID isolation. The researcher then visited each eligible patient, explained the study and obtained informed consent from those who were willing to participate. 

### 2.3. Data Collection

Semi-structured interviews were used to explore patients’ understanding of their fall risk in hospital. The lead researcher conducted all of the interviews using a pre-determined interview schedule (Appendix B). The interview schedule was generated by the research team, using an open-ended interview structure, as guided by Turner [33]. All interviews took place in participants’ hospital rooms of their corresponding wards during the weekday with efforts made to ensure minimal disruption to the participants’ clinical care and privacy. The interviews were audio-recorded via two dictaphones with the researcher taking field notes during and after the interviews. Demographic and relevant patient clinical information such as clinical fall risk status, was obtained by the lead researcher from patients’ medical records immediately after the interview. Data collection continued until the lead researcher was satisfied that sufficient information power was achieved [34]. Adequate information power depends on multiple variables such as the study aim, sample specificity, use of established theory, quality of dialogue and analysis [34]. Prior to data collection, it was estimated that approximately 15–20 interviews were needed for this research.

### 2.4. Position of the Researcher

Reflexivity involves reflecting throughout the research process on the researchers’ assumptions, expectations, choices and actions [35,36]. The lead researcher has a clinical background of physiotherapy practice (14 years) and nursing practice (3 years) and has cared for patients both in an acute and community setting. In disclosing her background to the participants, the lead researcher acknowledges that the patients may have an element of pride regarding their perceived capabilities and may not be as forthcoming with an ‘outsider’. This may influence the depth and quality of the recorded data; however, it also provides a valuable opportunity to situate the researcher within the hospital context. Researcher subjectivity and insights can be viewed as a successful component of reflexive thematic analysis [36].

### 2.5. Data Analysis

The theoretical flexibility of reflexive thematic analysis was best suited to address the research question in exploring both perceptions and patient behaviours in the context of a hospital admission [37]. The interviews were de-identified and transcribed by the lead researcher and subsequently compared to the audio-recording to confirm its accuracy. The mid- and post-interview field notes from the lead researcher were shared with the research team and reflected upon through debriefing sessions. The six-step reflexive process described by Braun and Clark [36], guided data analysis. To establish trustworthiness, the transcripts were independently coded by one researcher and co-verified by another. The research team independently analysed the coded data and created themes during a face-to-face meeting on the 27 September 2022, which resulted in highly comparable themes of the data. The major and minor themes were reviewed and refined until all authors were satisfied that the resultant analysis accurately represented their interpretation of the data. The findings section is presented under these headings with participant quotes labelled according to their assigned code, e.g., Participant 1 = P1.

## 3. Findings

### 3.1. Description of Participants

A total of 18 individual interviews were conducted from May 2022 to July 2022 at three regional public hospitals. These hospitals are from three separate local government areas (LGAs) with similar patient demographics and constitute a large 289-bed hospital, a 70-bed acute/subacute hospital and a 36 acute/subacute facility. There was an equal number of female and male participants (*n* = 9 each) with an average age of 69.8 years (SD ± 12.7, range 41 to 84 years). Additionally, two participants had mild cognitive impairment as per their clinical records, each returning a SMMSE score of 24 (codes: P11 and P15). The interviews were conducted in patients’ rooms in rehabilitation wards (*n* = 7), followed by orthopaedics (*n* = 5), acute/surgical (*n* = 3), specialised geriatric medical (*n* = 2) and a generalised medical ward (*n* = 1). The interview times ranged from 4 min 21 s to 22 min, 13 s (average time = 10 min, 10 s). Of the 18 participants, 67% had sustained a fall in the previous six months, with 28% of these falls occurring during their current hospital admission. Some of these resulted in major consequences with five participants experiencing a fracture from their fall at home, one experienced a significant head injury and two participants required a hospital admission for acopia. Two participants reported a history of multiple falls, which they attributed to their chronic health conditions. All participants, except for one [P4], were classified as having a high falls risk, according to their documented Falls Risk Assessment Tool (FRAT). Table 1 details the admission diagnosis for each participant.

Participants with chronic conditions such as Parkinson’s Disease, stroke, liver cirrhosis and cystic fibrosis had experienced multiple hospital admissions. These participants had good insight into their own health and in falls management. Participants with orthopaedic complaints, such as fractures, considered their condition to be temporary and were hopeful to return to their pre-morbid activities.

### 3.2. Description of Themes

From an initial analysis, the data codes were clustered into three broad patterns of meaning: Environment (extrinsic), Individual (intrinsic) and Outcomes. Eight minor themes were also derived from the data as subsets of the major themes. The first major theme (Environment) consisted of Safe mobilising in hospitals and Depending on others to prevent falls. The second major theme (Individual) comprised Insight into own needs, Confusion increases the risk and Fear of falling. The third major theme (Outcomes) consisted of Awareness of consequences, Falls education and Retraining the mind. These are displayed below in Figure 1. 

### 3.3. Theme One: Environment (Extrinsic)

#### 3.3.1. Safe Mobilising in Hospitals

Participants identified that hospitals can be a high-risk environment for falls. Hospital bathrooms were noted as the main concern, specifically the potential to fall in the shower. Slippery hospital floors were also highlighted because of the potential presence of water or bodily substances on these surfaces. One participant verified the need for suitable footwear, otherwise they would immediately slip over.

Gait aids were considered essential for mobility; however, some spoke of their physical limitations with using such equipment, especially if they were not in close proximity. An organised hospital layout contributed to feelings of safety in one participant,

“*They’ve got to keep everything in its place. So that makes me feel safe*”[P15]

#### 3.3.2. Depending on Others to Prevent Falls

Extrinsic influences such as nursing staff supervision and call buzzer technology were considered to be the main component to falls prevention in hospital. The assistance provided by nursing staff eliminated the need to engage in risk-taking behaviours, 

“*If I ask for assistance it comes, so I don’t have to take any risks*”[P1]

The reliance on nursing staff for transfers and personal hygiene/toileting was clearly voiced with most acknowledging that they would not hesitate to seek assistance. Frustrations were expressed with slow or unanswered buzzers, 

“*There are times when I’m ringing for a bell. I’m ringing the red lights and no one comes and I want to use the bottle. They don’t come and I’m getting desperate*”[P7]

This demonstrates the vulnerability of patients and their reliance on others for their personal needs and dignity. Despite their obvious distress, participants were quick to defend the nurses and discussed their heavy workloads and patient priorities. They recognised the low staffing issues combined with the high acuity of some patients. The fact they were able to empathise with the nurses’ schedules demonstrates their insight into the realities of hospital wards. Some people were also wary of using their call buzzer and cited altruistic reasons for not wanting to seek assistance, 

“*I don’t expect them to drop everything for me. I just find I need help going to the toilet. I know they’re busy. When you want to get back to bed, I buzz and then I say to myself, ‘You’re being selfish’*”[P6]

Despite these misgivings, most participants would seek assistance as applicable because they appreciated the importance of staying safe. Additionally, one participant spoke about the presence of falls alert signs as reminders for additional monitoring by staff. The person elaborated on possible ideas for the future, 

“*There needs to be some kind of collar, like an electronic tracking bracelet that they would put on the patients and nurses at the front desk who’ll be monitoring those bracelets to see where the patients are, once they bypass the sensor from their door from the room*”[P5]

Again, this supports the assertion that extrinsic influences such as hospital staff and technology, are considered essential to staying safe in hospital. 

Although there is a reliance on nursing staff, participants asserted that patients needed to have accountability for their choices and behaviours in hospital. As a participant explained,

“*I think the patients have gotta take responsibility to some extent. You can’t just come in and be pandered over … patients gotta take responsibility for what they can do and what they can’t do. They can ask for assistance. That’s why we’ve got the bell*”[P8]

This reinforces the importance of including the person as a ‘partner’ in falls prevention and safety management in hospital. 

### 3.4. Theme Two: Individual (Intrinsic)

#### 3.4.1. Insight into Own Needs

Despite their past medical histories or current circumstances, some participants did not consider themselves to be at risk of falling in hospital. This was mostly attributed to their own personal behaviour and willingness to seek assistance, 

“*‘cause there’s help always and I won’t do anything unless I’ve got help with me. I won’t try and do anything that I know I’m not gonna be able to do*”[P16]

“*Am I at risk of falling? Not as long as I do what I’m told*”[P7]

Only a few participants identified their wish to retain their independence. Risk-taking behaviour was accepted as part of their recovery, 

“*I do it myself, I just … there’s no other, no other way to do it. I’m gonna have to do it at home*”[P18]

When participants were asked “*why do patients fall in hospital?*”, most responses attributed the blame to the individual, rather than to the hospital environment. A variety of mechanical reasons for falling in hospital were revealed, such as balance difficulties, weakness or unsteadiness on one’s feet. One participant who had experienced numerous hospital admissions offered, 

“*I’m noticing a lot with old people they’ll get up out of their bed. They might be gasping for oxygen and they’ll stand up too quickly out of bed. They’ll try and catch themselves and then they’ll start wandering or calling out and they’ll lose blood pressure very quickly and they’ll fall over. I’ve seen this happen a couple of times*”[P5]

This participant demonstrated a high level of insight, as they understood that falls could be attributed to medical conditions and not just mechanical causes. 

Participants also recognised that poor insight into their own needs could lead to falls in hospital. As one related,

“*Not concentrating, I believe. Or getting their ambitions and their capabilities mixed up. Just thinking they’re better than they really are at the time*”[P14]

#### 3.4.2. Confusion Increases the Risk

Confused patients or those with dementia have an increased falls risk in hospital, according to participants. One person spoke about their recollections as a patient, 

“*I’ve seen elderly patients get out of bed and wear bed pans as their shoes ‘cause they thought they were their shoes. I’ve seen dementia people walk around and just fall over*”[P5]

Some acknowledged the unfamiliarity of hospitals compared to their usual surroundings, which could lead to disorientation. However, this was not limited to an unfamiliar hospital environment, as one participant spoke of their disorientation at home in the middle of the night, which led to multiple falls. The two participants with mild cognitive deficits provided their perspectives of their falls risk in hospital. One had good insight into their potential to fall, 

“*If I’m not look well, not watched and guarded, probably yeah I could (fall)*”[P11]

In comparison, another person with mild cognitive deficits reported feeling “*safe*” in hospital. However, this person went on to describe their feelings of disorientation during their admission, 

“*It was a small case of fright, I didn’t know where the heck I was! I didn’t know anything*”[P15]

#### 3.4.3. Fear of Falling

Interestingly, the fear of falling construct surfaced amongst many participants. One person spoke about their concerns, 

“*I’ve just had a massive fall at home less than a week ago and I’m just absolutely terrified of falling again. This is where I make sure I’ve got one or two people with me when I get up and go to the toilet and back*”[P9]

This also indicates dependency on hospital staff in maintaining safety. Prior to their hospital admission, this person [P9] was fiercely independent and entry to a nursing home was considered unlikely. The psychological impact of a fall can impede a person’s recovery and ability to remain independent; however, a ‘fear of falling’ does not always develop after a fall. As demonstrated in one participant, 

“*I have never had a fall, that’s why I’m so paranoid I think. I just don’t want to it’s the last thing I want to do*”[P12]

This participant [P12] disclosed that they had witnessed the outcomes from falls in other family members, which led to dependence on others and decreased participation in society.

### 3.5. Theme Three: Outcomes

#### 3.5.1. Awareness of Consequences

Participants demonstrated awareness of the possible consequences of falling. They correctly identified the likelihood of soft tissue injuries, fractures, head trauma and even death as potential outcomes. As one participant noted,

“*They break their hip, get pneumonia and then they’re on their way out. They just don’t get better and better and it’s all part of the grand scheme of death basically*”[P5]

They recognised the implications of these consequences for their future, thus appreciated the significance of falls prevention. Unfortunately, some of their awareness came from falls experience. Participants who had previously fallen, reflected on their experiences and shared their lessons learnt. These experiences shaped their beliefs and subsequently altered their approach to falls prevention. One patient who had a stroke explained, 

“*After having my falls, it makes you concentrate more. ‘Cause it’s scary being on the floor, flopping around like a dying fish. So that’s what keeps you a lot more focused if you have had a fall*”[P16]

#### 3.5.2. Falls Education

Participants valued the expertise of allied health professionals and appreciated the falls education they received. They understood the role of the health professional as key to their recovery, 

“*I figured that they know what they’re doing, so if you want to get better, you listen and do*”[P3]

Interestingly, many participants denied that they had received falls prevention education from hospital staff. In particular, the two people with mild cognitive deficits replied,

“*No. Be nice if they did” [P11], with another commenting, “Not that I know of or remember. They may have, but I don’t remember*”[P15]

Field notes depicted both patients as wearing orange grip socks, call buzzer nearby and their gait aid within reach. These participants may not have been informed that these strategies were part of their fall prevention plan, or their mild cognitive deficits may contribute to poor recollection. 

A few participants discussed their frustrations about receiving mixed messages and conflicting advice from health professionals. Feelings of “*not being listened to*” and poor communication were disclosed, including with family members. Some advocated for improved patient/clinician communication, with one even suggesting,

“*I always say to everybody to ring my wife first to keep her in the loop ‘cause when you get old like us they seem to think that your brain is gone and you can’t think for yourself*”[P7]

Participants were receptive to falls education with some planning to attend outpatient falls prevention programs upon discharge. As one person noted, 

“*If I survive this ordeal and come out of hospital, I’ll have to do something about it because it seems like simple little falls are gonna be a tragedy to me*”[P7]

#### 3.5.3. Retraining the Mind

Concentration at all times and remaining “*free of distractions*” was considered essential to safe mobility in both hospital and at home. One person articulated their difficulty in adhering to falls prevention strategies, 

“*Sometimes I forget and I’ll start to head off and then the alarm goes and I realise I’m not supposed to stand up. I’ve spent 70 years doing [things] how I want to and I’ve been here a month it takes a long time to get over old habits. You know but I’m gonna have to learn because I can’t afford to have too many more falls*”[P2]

There is a need for rehabilitation to prevent further falls, though participants recognised that change would not be instantaneous. Retraining the mind was considered essential to help participants adjust for their future, 

“*Your mind is not [the] same as your body. Your mind is still strong, you know? But your body, it doesn’t give the same amount, so you have to work it out. What you can do and what you can’t do. You have to teach yourself, train yourself again to be really strong*”[P10]

An insightful participant spoke about their need to consider their “*future self*” above their pride. Participants remained realistic but hopeful that improvements could be attained for their future and they recognised their role in their recovery. Positive attitudes, hope and accepting responsibility were highlighted as the driving contributors to facilitate adaptations for their future selves. 

## 4. Discussion

A reflexive thematic analysis was conducted to understand patients’ perceptions of their fall risk in rural and regional public hospitals in Victoria, Australia. Recent statistics illustrate the high incidence of reported falls in these areas compared to metropolitan regions [8], which highlights the importance of this research. The findings indicate that perceived falls risk in rural and regional Victorian public hospitals can be represented by an interplay of the hospital environment, individual intrinsic factors and outcomes. This conceptual relationship between the three major themes demonstrates the accumulative effects of perceived falls risk and is characteristic of the multifactorial nature of falls prevention. 

Participants recognised the high-risk nature of the hospital environment and understood the potential outcomes following a fall; this included those individuals with mild cognitive impairment. They acknowledged an interplay of both intrinsic and extrinsic factors are vital to fall prevention management. Interestingly, there were no reported differences across wards despite variations in patient acuity. The roles of the hospital staff, technology and falls education were important for patient safety but participants also acknowledged their own accountability and responsibilities in staying safe in hospital. Similar to Gettens, Fulbrook, Jessup and Choy [23], participants had confidence in the nursing staff to keep them safe, but also expressed frustration with delayed responses, which may lead to risk-taking behaviour [24]. Comparably, delays in responses to call buzzers discourages patients from seeking assistance [38]. In the theme “Depending on others to prevent falls”, participants maintained that they would continue to seek assistance from nursing staff, despite barriers, because they recognised the importance of staying safe. Hospital staff can be guided to provide appropriate education and maintain open communication with patients about the importance of utilising call buzzer technology.

An interesting finding from our analysis is that most participants did not consider themselves to be at risk of falling but were willing to comply with nursing staff and seek assistance. Participants demonstrated awareness of their physical limitations and knowledge of the consequences of falling. Previous qualitative studies show that patients’ self-perceived fall risk differed to their clinical risk [19,38,39]. Theoretically, participants may not have considered themselves to be at risk of falling, given their accurate insight into their physical capabilities and compliance with fall prevention strategies. In Dolan et al. [40], participants used the term “balance problem” instead of “fall risk”, which may have originated from their perceived high self-efficacy. It is also possible that the self-esteem and self-identity of people may be associated with others perceiving them to be independent, rather than as “fallers” [41]. Self-identity can shape a person’s response to fall prevention and management [42] and attitudes can be either an enabler or a barrier to falls prevention [22]. Given the above information, future studies could explore the impact of changing terminology from “falls risk” to a more progressive term like “falls aware”. Studies have found that removing fall risk screening tools have resulted in favourable outcomes [43,44]. Terms such as “falls aware” instead of “fall risk” may have positive implications for person-centred falls education and facilitate the patient/health professional dyad for collaboration on fall prevention strategies.

The ‘fear of falling’ concept, otherwise known as post-fall syndrome, relates to an anxiety of falling even though they may not have experienced a fall [45,46]. This was the case in our findings, in which two participants from our study disclosed their fear of falling: one from a lived experience perspective and the other without a falls history. Fear of falling is associated with increased anxiety, reduced confidence and self-efficacy, reduced self-rated quality of life, a higher dependency with activities of daily living and a higher risk for falling [47,48,49]. Because of the consequences associated with fear of falling, health professionals need to be able to recognise this in patients and tailor falls prevention strategies accordingly. The updated World Falls Guidelines strongly recommend including an evaluation about the concerns or fear of falling in older adults as part of a multifactorial risk assessment [16]. Interestingly, these recommendations also acknowledged the importance of language, with an older adult panel favouring the term ‘concern’ over ‘fear’ [16]. As highlighted earlier, health professionals should consider the impact of language when communicating with inpatients and incorporate the preferred terminology. The preferred terminology should also be reflected in healthcare policies and education.

There is no single, definitive strategy for falls prevention, although a meta-analysis found that falls education reduces inpatient falls rates [50]. The provision of individualised falls education increased awareness and empowered patients with the skills to engage with falls prevention strategies [22]. One study found that many cognitively intact participants were not able to recall the falls prevention education they received on admission [39]. These findings are comparable with our study in which many participants, including those with mild cognitive deficits, could not recall receiving falls education. From a lived experience perspective, Mion [51] suggested that health personnel should reiterate fall prevention strategies with every patient encounter, despite cognitive status, because of rapid alterations in clinical environments. 

Evidence suggests that effective clinical communication requires active participation of patients and their families/or carers, resulting in improved outcomes [52]. The involvement of family and/or caregivers with falls prevention was scarcely mentioned by participants in this study. It is possible that COVID-19 restrictions imposed on family members and visitors to Victorian hospitals may have impacted family involvement. Instead, personal accountability, including ‘retraining the mind’ was considered crucial to staying safe and adapting to an unknown future. Future falls prevention education in hospital should focus on engaging the person in their own falls management and empower patients with the tools/strategies to help them adjust to their temporary or permanent clinical circumstances.

### Strengths and Limitations

Notably, people with cognitive impairment are often omitted from research [32], yet studies show that this cohort has an increased risk of falling [13,53,54]. The inclusion of people with mild cognitive impairment in this study provided representation of falls risk perception in this cohort. Only two people were identified with mild cognitive impairment in this study, so readers should be mindful of this potential limitation. Similarly, this study included the falls risk perception of adults aged 40 years and over. Guidelines specify that people aged 65 years and older have the highest risk of falling in hospital, along with people aged 50 to 64 years with an underlying condition [2]. As a result, falls research is usually tailored towards this population. Two of the participants who had previously fallen (P5 and P16) were aged between 40 and 50 years of age and had experienced numerous hospital admissions from their complex medical histories (cystic fibrosis and stroke). The falls risk insights provided from these participants contributed considerably to the findings from this study. Therefore, it is worthwhile for falls researchers to consider including this age group in future studies, especially for those participants with underlying medical conditions. The age range selected for this study was established from a clinical audit from one of the participating hospitals; however, future studies could include younger adults for which the data may provide further understanding of this issue.

This study was conducted in rural and regional Victoria, Australia, across three public hospitals. Therefore, the findings may not be applicable to metropolitan areas and/or private hospitals. The study is also limited by the exclusion of people with severe cognitive deficits and those with difficulties understanding English. This decision was made for feasibility and ethical purposes in that we were unable to have additional people, e.g., family members or interpreters in the hospital room due to COVID-19 limitations. Some of the individual interviews were interrupted due to routine hospital procedures, which may have disrupted the flow of the interview and contributed to shorter interview times. Lack of privacy may also have been a limiting factor as some participants were in a shared hospital room, which may have inhibited their willingness to speak freely. All efforts were made to ensure the participants’ privacy and comfort. Despite these limitations, the researchers consider their analysis to be an accurate representation of the data. 

## 5. Conclusions

Hospital falls prevention and management continues to be investigated on a global scale with recent efforts focused upon understanding the patients’ perspectives of their falls risk. This research is important because despite falls education some patients still engage in risk-taking behaviour. Our analysis showed that participants recognised the hazardous nature of a hospital and identified the need for both intrinsic and extrinsic factors to stay safe. Falls risk perception in rural and regional Victorian public hospitals can be conceptualised as an interplay of the hospital environment, individual intrinsic factors and outcomes. Falls education needs to be delivered frequently, with the focus on empowering the patient. Future studies could explore the influence and diversification of language used by health professionals in the delivery of falls education and consider the use of alternative terminology. Our findings originate from a rural/regional public hospital context; however, future research could examine these issues in alternative settings. 

## Figures and Tables

**Figure 1 ijerph-20-00454-f001:**
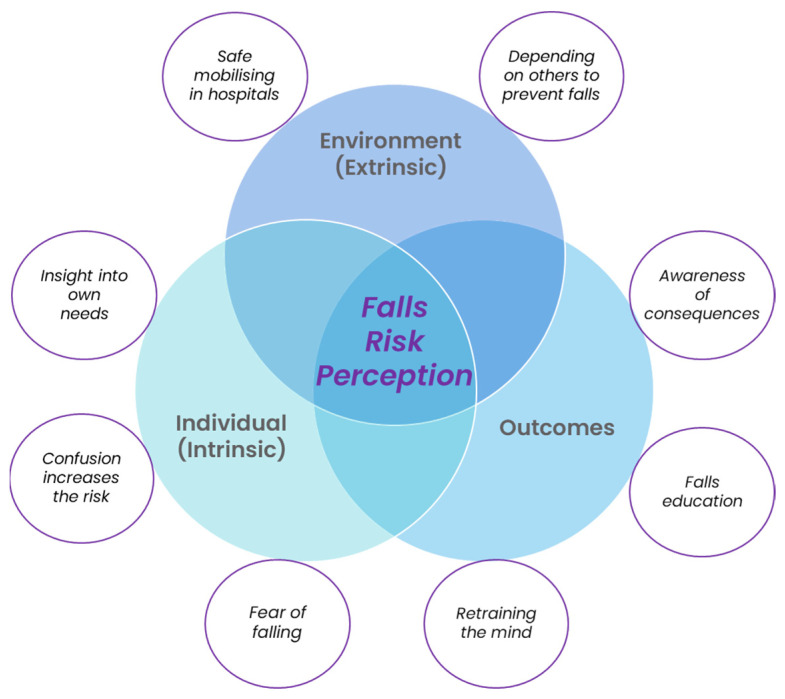
Thematic Model.

**Table 1 ijerph-20-00454-t001:** Admission diagnosis for each participant (Code: P1 = Participant 1, P2 = Participant 2).

Admission Diagnosis for Each Participant			
P1	Subdural haemorrhage	P10	Orthopaedic complications post knee surgery
P2	Exacerbation of Parkinson’s Disease	P11	Cardiac complaint
P3	Fractured pelvis and shoulder	P12	Uncontrolled peristomal leakage
P4	Fractured hip	P13	Unstable angina
P5	End of life care: cystic fibrosis	P14	Toe amputation
P6	Neuropathy causing multiple falls	P15	Fractured tibia and fibula
P7	Syncope and falls	P16	Fractured hip (Past history: Stroke)
P8	End stage liver cirrhosis	P17	Gait training post transtibial amputation
P9	Fall at home	P18	Fractured ankle

## Data Availability

The data presented in this study are available on request from the corresponding author. The data are not publicly available due to privacy/ethical reasons.

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
