# Peer review of "Exploring Hospital Inpatients’ Awareness of Their Falls Risk: A Qualitative Exploratory Study"

_ijerph, 2022, doi:10.3390/ijerph20010454_

Round 1
Reviewer 1 Report
It has been a pleasure to review your article.
Some minor changes will favor a little the understanding of the manuscript.
The reasons why the interviewees are afraid are situations that can happen to another group of younger people. Perhaps the perception of a possible fall in a bathroom or on a wet floor is different, but the fact of falling is the same.
In future investigations, I suggest that they include younger people since the data can shed another, clearer interpretation for the elaboration of the results.
Change in keywords. I recommend not using the same terms in the title and keywords. What originates is a loss of power when searching for the article.
The origin or starting point is undoubtedly very important. We are talking about hospital falls. Areas that at first should not have this type of event happen, but we see or at least that is how it is reflected, it continues to happen.
In material and method, it is commented that "The interviews were carried out in the hospital rooms of the participants during the working day and efforts were made to guarantee the minimum interruption of clinical care and the privacy of the participants"
However, it is later stated that "Most of the interviews were conducted in rehabilitation rooms (n = 7), followed by orthopedics (n = 5), acute/surgical (n = 3), 146 medical specialists in geriatrics (n = 2) and a general medicine ward (n = 1)."
This point must be clarified—the place where the interviews were conducted.
Another value that must be taken into consideration is the low number of interviews with participants with mild cognitive impairment. The result obtained is highly biased by the performance of only two individuals.
Other data that can be exposed is how they have the same number of men as women, it is to differentiate the responses by sex. Kas women are more likely to fall due to osteoporosis and this issue is not questioned in the manuscript. How does the fear of falls affect the female population?
I would like to see in the conclusions and in view of the reasons why there are fears of falls, the possible solutions that can be given, not only the education on falls that should be given to any patient and family members who are in a hospital. but the infrastructure mechanisms that must be modified to avoid a high percentage of falls
Reviewer 2 Report
Comments to Author:
Ms. Ref. No.: ijerph-2083322
Title:
Exploring hospital inpatients’ awareness of their fall risk: A qualitative descriptive study
The authors reported a study that aimed to explore the perceptions and experiences that influence a patient's understanding of their fall risk in regional Australian hospitals. A total of 18 inpatients from three rural and regional Victorian public hospitals were interviewed. The authors summarized three major themes: environment (extrinsic), individual (intrinsic), and outcomes. Some interesting findings were discussed. However, based on the current version, the descriptions of some critical points are inadequate or completely missing.
ABSTRACT:
1. It is not suggested to add citation(s) in the Abstract
2. Readers are interested in more details about the participants, e.g., demographics.
INTRODUCTION
1. The current version didn't clearly justify the background of why it's important to study fall-related perceptions in hospitals. Is it different from falls at home? Why hospitals?
2. line 30: please add detailed numbers/rates about hospital fall rates, rather than "remained largely unchanged"
3. line 35: it is confusing to see that the authors mentioned three domains from WHO but only illustrated "safer people". How about the other two domains, i.e., safer environments and safer policies and legislation?
4. line 62: what are the differences between other hospitals and the ones in the Victoria setting? any specific reasons? If yes, please add the details.
5. again, it is not clear about conducting a new study about falls in hospitals. The research gap(s) is(are) not sufficient.
RECRUITMENT:
1. line 95-96: did the authors intentionally recruit MCI/dementia patients or not? eligible if SMMSE >18, but the authors mentioned "... why the researchers wanted to include people with mild to moderate cognitive impairment"; what are the criteria for mild/moderate CI?
DATA COLLECTION:
1. line 114-115: it is too subjective to say, "data collection continued until the lead researcher was satisfied...." what is sufficient information power?
POSITION OF THE RESEARCHER
1. line 119: what is the clinical background of physiotherapy of the researcher? 5 years of working experience? or?
FINDINGS
1. line 142: more details should be added about the three hospitals. any similarity? any differences?
2. line 144: similar to the previous comment, i.e., how to define MCI?
3. line 149-150: more details about the fall history should be added, e.g., how many falls/what was the consequence of the previous fall... The history may affect the participants' perceptions
4. line 152: what are the falls risk assessments? please clearly record and display the findings (if any)
5. Table 1: is there any reason to bold the P1 and P10?
DESCRIPTION OF THEMES
1. line 171: this sentence is a description of Methods, not Results
2. line 175: the sub-theme "safety in hospitals" is too general. the authors are strongly advised to narrow it down, e.g., balance/gait safety
3. line 185-204: please present the findings of "depending on others to stay safe" related to falls
4. line 190-192: why use the bold font? any specific reasons?
DISCUSSION/STRENGTHS AND LIMITATIONS
In general, the sample was very limited, e.g., only having two MCIs. The authors cannot jump to a conclusion like line 420 "... provides further understanding of falls risk perception in this cohort (MCI)". Also, the authors may forget to emphasize the importance of studying falls in hospitals. There are many publications talking about falls. Why is it important to have a new qualitative study on fall perceptions? Furthermore, the participants were recruited and interviewed in different wards, which may significantly affect the findings.
Reviewer 3 Report
Comments and suggestions for the authors are provided in the attached file.
